# Transparency and Disclosure and Financial Distress of Non-Financial Firms in India under Competition: Investors' Perspective

Jagjeevan Kanoujiya [1], Rebecca Abraham [2,*], Shailesh Rastogi [1] and Venkata Mrudula Bhimavarapu [3]

1 Symbiosis Institute of Business Management, Symbiosis International (Deemed University), Pune 412115, India
2 Huizenga College of Business, Nova South Eastern University, 3301 College Avenue, Fort Lauderdale, FL 33314, USA
3 Symbiosis School of Banking and Finance, Symbiosis International (Deemed University), Pune 412115, India
* Correspondence: abraham@nova.edu

**Abstract:** Transparency and disclosure (T&D) of information trigger the interest of all stakeholders, including investors in a company. Cognizance of the company's financial health before investing is very necessary. Disclosure of information in the firm's financial reports reflects the firm's financial performance. A firm's financial health protects investors' and other stakeholders' interests and the firm's long-term sustainability. Owing to the importance of T&D and a firm's financial health, this paper investigates the impact of T&D on the financial distress (FD) of non-financial firms (NFFs) listed in India. This study examines both linear and nonlinear connectivity of T&D and financial distress (FD). Their association is also investigated in a competitive scenario (under the moderating effect of competition). The panel data analysis is incorporated into the study having 78 NFFs as cross-sectional units with a timeframe from 2016 to 2020. Altman Z-score measures a firm's FD (higher Z-score means low FD). BOS (Berger, Ofek and Swary) and AC (Almeida and Campello) scores are taken to consider investors' perspectives of the firm's FD. The T&D and Lerner indexes are used to assess the level of T&D and competition. The findings reveal that a higher T&D level decreases a firm's financial stability or increases a firm's FD. In nonlinear association, it is found that T&D has an inverted U-curved connection with financial stability or U-curved association with FD. It indicates that initially, higher T&D reduces FD, and after a threshold, it increases FD. However, under competition, T&D is not found to be significantly impactful for FD. The study is novel as no previous study has focused on such association under competition and taking investors' perspective of a firm's FD.

**Keywords:** corporate governance; transparency and disclosure; financial distress; competition; investors

**JEL Classification:** G30; G32; G33; G38

## 1. Introduction

Pertinent financial information is essential for all firms' stakeholders to make sound economic or financial decisions. Information regarding the firm's financial distress condition is necessary for concerned stakeholders because financial distress may lead to the failure of the firm's business activities. Companies try to keep the information as virtuous as possible so unwanted details cannot be publicised. Therefore, transparency and sufficiency in information disclosure are essential to good corporate governance. For good governance mechanisms, certain steps are incorporated to improve the financial reporting standards, including transparency in information disclosure and widening voluntary disclosures. Good corporate governance practice is also associated with the principal-agent

issue (Jensen and Meckling 1976). Managers are agents of the firm, while shareholders are principals. Jensen and Meckling (1976) posit that conflicts between principals and agents affect firm performance. Studies such as Ross (1973), Fama and Jensen (1983) and Mallin (2016) advocate that a suitable corporate governance mechanism helps in minimising the issue of principal-agent conflicts.

The global blitz of the financial crisis of 2008 and the drastic financial transgression in Lehman Brothers, Enron, AIG, World COM, and others have again created a flutter among the stakeholders. It has redirected researchers, regulatory bodies, investors, and policymakers to explore the extent of governance practices in corporate and their impact on financial stability (Shahwan 2015). Generally, corporate governance obtains its quality check from the T&D of the firm's information, for instance, shareholders relations, ownership, control structure, the board of directors' features, policy, and compliance.

The available literature on corporate governance and financial performance has inconclusive views on the association of corporate governance and firm performance. For instance, Younas et al. (2021), Hodgson et al. (2011), Black et al. (2006), Huang (2010), and Varshney et al. (2015) advocate that suitable governance mechanisms reinforce firm performance; hence improving financial stability by increasing profitability. However, Wahba (2015), Hambrick and D'Aveni (1992), and Daily and Dalton (1994) indicate a negative connection between CG to financial performance. Omran et al. (2008) and Makki and Lodhi (2014) have indicated no significant connection between governance and financial performance. However, T&D as an essential element of governance and its connectivity to the firm's financial distress are not much explored.

In the Indian context, the nation's economy has experienced substantial regulatory and environmental transitions to participate in the global economy. Currently, many reforms are also realised in the execution of good governance practices under the Company Act 2013 (Thapar and Sharma 2017). Transparency in information disclosures is also felt essential after the scams of eminent business failures like UTI scams, Satyam, scam of Ketan Parikh, and stock market fraud due to bad governance practices in corporate (Thapar and Sharma 2017). To counter these frauds and other business failures in future, the Indian government has taken steps such as the Company Act 2013 and the new Insolvency and Bankruptcy Code (IBC) 2016. Indian corporate sector realises improvements through these reforms. However, there are still overreaching issues, as evident from recent corporate failures like Kingfisher, Yes Bank, Jet Airways, Bhusan steels, and many others (Balasubramanian 2013).

Although significant steps are taken regarding improving CG practices and promoting financial stability, there still exists controversial views on the impact of T&D (as an inseparable part of corporate governance (Jatiningrum et al. 2023)) on a firm's FD, particularly in developing nations like India. The governance mechanisms in India's corporate sector are transforming from a conventional regulatory model to a market-oriented one. Hence, market competition also has an essential role in a firm's FD. Looking for the impact of corporate governance on a firm's FD in such a competitive environment is an important aspect that needs to be investigated.

In essence, FD emits a negative signal about the firm's financial performance. Good governance practices encourage truthfulness in assessing the distressed firm's financial position, while T&D requires publicising such adverse financial results. Can the positive signals from CG considering T&D mitigate the negative signal from FD?

From the above discussion, it is observed that T&D is essential for good governance. However, its role in determining a firm's financial health is not much explored. Moreover, studies on T&D and firms' performance are available only in developed nations. It is also observed that developing countries such as India have witnessed several critical regulatory reforms in response to corporate failures. However, studies on T&D concerning a firm's financial distress are rarely conducted in the Indian context. Moreover, the existing evidence on T&D and FD nexus has inconclusive views. Therefore, this study fills this research gap by providing novel evidence on the association of T&D with FD of non-financial firms in India.

Thus, the present study adds to corporate governance and FD literature by exploring the T&D, competition, and FD level of 78 NFFs listed in BSE100 (India). Furthermore, the following objectives are set to contribute to the existing body of knowledge:

- To examine the impact of T&D on the firm's FD of listed NFFs in India.
- To examine the impact of T&D on a firm's FD under the influence of competition in NFFs in India.

As T&D is a significant factor for financial distress in several ways (linearly, nonlinearly, and under the interaction of competition), the study's findings significantly contribute to the existing literature by providing strong evidence with its novel approach. This approach uses multi-methods to look for the relationship between T&D and FD from different angles. As the study also considers the investors' perspective of such a relationship, it entrusts prudential implications for investors' decision-making to consider the T&D of a company as a critical element for its long-term feasibility. Moreover, this study also brings notable implications for policymakers and managers to take T&D as a serious matter of concern for financial distress.

The remaining part of the paper includes Section 2 for the literature review, Section 3 for hypothesis development, and Section 4 for data and methodology. Sections 5 and 6 describe the results and discussion, respectively. Finally, Section 6 concludes the paper.

## 2. Literature Review

This paper aims to find the impact of T&D on a firm's FD. This section reviews the extant literature related to the study's objectives. First, it reviews the literature on corporate governance concerning firm performance. Second, it looks for studies related to T&D and financial performance. Third and last, it builds the hypotheses for the empirical test.

### 2.1. Financial Reporting and Its Importance

Corporations need to follow a more transparent, disclosed, and consistent approach to information reporting, especially financial information, to gain market share (Hussin and Othman 2012; Cheffins 2013). This approach helps the investor to act in a more sophisticated way for decision-making. With the help of information technology, most businesses operate globally. Thus, internationally acknowledged reporting frameworks of corporations' information (specifically financial information) must be followed. Before 2005, corporations only followed the only option "Generally Accepted Accounting Principles framed by International Accounting Standard Board". After that, the "Financial Accounting Standard Board framed International Financial Reporting Standards (IFRS)" to ensure a sound and transparent mechanism for disclosing corporations' financial performance (Hussin and Othman 2012; Cheffins 2013). This situation may result not only in more disclosed and transparent governance practices at corporations but could also clearly show the obligations of an excellent corporate person. In the Indian context, corporations follow "Indian Accounting standards" primarily based on IFRS. The regulatory measures and technological reforms have also substantially changed business and information delivery (Bhimavarapu et al. 2023).

### 2.2. Corporate Governance and Firms' Performance

Many studies empirically investigate the association of corporate governance (CG) with firm performance. As per Cheffins (2013), the first such study is potentially demonstrated in the 1970s. Solomon (2020) finds that the nature of governance cannot be defined in general because it varies depending on the governance code, cultural environment, frameworks, policymakers, and researchers of the concerned economy (Armstrong and Sweeney 2002; Solomon 2020). King IV defines corporate governance as "the practice of ethical and effective leadership by the governing body towards the achievement of the following governance outcomes: (i) Moral culture (ii) Good performance (iii) Effective control (iv) Legitimacy". Good governance induces the firm to hold higher performance standards

(Bebchuk et al. 2009), while firms with inadequate corporate governance mechanisms face lower firm performance standards (Gompers et al. 2003).

Rajagopalan and Zhang (2009) advocate that the betterment of the governance mechanism boosts the shareholders' confidence, which improves their investment in firms and the firm's financial stability. Durnev and Kim (2005) indicate that companies with a higher potential for growth and higher requirements for more financing (external) show better governance and disclosures, particularly in countries with weak legal protection. Berglöf and Pajuste (2005) study the factors associated with T&D in post-transition economies. In their cross-country study, firms having more controlling owners, large firm size, less leverage, slower growth, and higher growth market-book ratio have higher T&D levels. However, contrary to Durnev and Kim (2005), they do not find evidence supporting the hypothesis that firms needing more financing (external) have more T&D levels.

The existing studies on corporate governance concerning a firm's performance still have contrasting views. As per Younas et al. (2021), Hodgson et al. (2011), Black et al. (2006), Huang (2010), Varshney et al. (2015), and Bai et al. (2023), suitable governance mechanisms reinforce firm performance. Therefore, these mechanisms safeguard firms against the threat of FD (Parker et al. 2002; Wang and Deng 2006; Abdullah 2006; Li et al. 2008). However, Wahba (2015), Hambrick and D'Aveni (1992), and Daily and Dalton (1994) indicate a negative connection between governance to financial performance, which implies that higher corporate governance downgrades financial stability. Omran et al. (2008), Makki and Lodhi (2014), and Rastogi and Kanoujiya (2022) have indicated no significant connection between governance and a firm's financial performance. Hence, the relationship between corporate governance and financial stability is inconclusive. However, governance practices regarding T&D and their connectivity to the firm's financial distress are not much explored.

### 2.3. Transparency and Disclosure and Firm's Performance

The dramatic breakdowns in the USA's big corporate names, such as Arthur Anderson in 1998 and Enron in 2001 and a similar kind of dramatic collapse of Marconi in the UK (Cheffins 2013), have accelerated the concerns for transparency in good corporate governance. Abdul-Qadir and Kwanbo (2012) and Hussin and Othman (2012) evince that corporate governance gets continuous importance due to the dramatic failure of many big companies. This situation includes AIG, HealthSouth, Word.com, the Lehman Brothers in the US and Megan Media, Transmile and NasionCom in Malaysia. Moreover, the dramatic liquidation of 26 banks in Nigeria with fraudulent financial reporting in the global financial crisis is another example in the long list of 2008 (CBN 2010). Furthermore, Satyam's collapse in India is considered among the top 10 financial scandals in the world's corporate history. The failure of suitable governance mechanisms and lack of transparency are the main loopholes in such dramatic breakdowns (Hussin and Othman 2012; Abdul-Qadir and Kwanbo 2012; Cheffins 2013). Jatiningrum et al. (2023) and Bhimavarapu et al. (2023) suggest that T&D of information is essential for good governance practices.

According to Chau and Gray (2002) and Amba (2014), firms reflecting increasing returns capture investors' interest; therefore, certain companies intentionally forego T&D and try to hide losses only to gain shareholders' confidence. On the other hand, firms with more transparency and disclosure of information result in more financial returns. Moreover, many studies indicate a trade-off association between T&D and a firm's financial performance (Watson et al. 2002; Wallace and Naser 1995; Clatworthy and Jones 2006; Ball et al. 2003). Sun et al. (2023) indicate that disclosures of integrated information on environmental, social, and governance negatively impacts a firm's value. In contrast, Ahmed and Courtis (1999) and Akhtaruddin (2005) find no significant association of CG's influence on T&D with a firm's financial performance. In this regard, Peters and Bagshaw (2014) advocate that governance's influence on T&D is not always an impacting factor for a firm's financial performance. Rastogi and Kanoujiya (2022) also identified a similar outcome for Indian banks. Other significant factors might be responsible for the firm's

higher financial performance, such as sales growth, technology, and capital returns. It is observed from the literature that numerous studies are available on corporate governance concerning financial performance. However, T&D as an essential element of governance is explored less. In addition, the studies exploring the relationship between T&D and financial performance have inconclusive views. Furthermore, the direct impact of T&D on a firm's financial distress is rarely investigated. Moreover, a large chunk of studies concentrated on developed economies. Therefore, this study aims to find the nexus between the two in Indian firms to fill the existing research gap.

*2.4. Hypothesis Development*

The technological enhancements in the last few years have created a revolutionary change in information delivery systems. Therefore, the T&D of information by a corporate is essential to know the good governance and performance of the corporation. This situation has also led to competitive challenges for the corporates to survive in the market. Hence, it is essential to understand how T&D impacts a firm's financial distress.

The existing literature is enormous and that puts light on the issue of transparency and disclosure of information and its associated costs and benefits for firms and their shareholders. The higher T&D levels benefit the firm's valuation only under certain conditions. A few essential conditions are: (1) lower investors' uncertainty (Hail 2002; Durnev and Kim 2005); (2) enhanced market-level interest (Lang et al. 2012); (3) better protection of investors' rights (Östberg 2006; Bebchuk et al. 2009); and (4) lower cost of capital (Botosan 2006; Frost et al. 2005). Lai et al. (2014) advocate that a higher degree of disclosure lowers the problem of information asymmetry, which brings on the management to perform in the shareholders' best interest. This situation improves the overall efficiency of the investments in the capital market and enhances the firm's financial stability. Chau and Gray (2002) and Amba (2014) also suggest that higher transparency and disclosure improve financial stability.

However, James-Overheu and Cotter (2009), in their study on Australian firms, found evidence that T&D of information fails to reduce a firm's financial distress. In support of Hermalin and Weisbach (2012), they argue that a higher level of T&D puts gratuitous pressure on managers who are monitored by the market; hence, they cannot give their best. It results in enhanced financial distress. Similarly, Berglöf and Pajuste (2005) suggest that a higher degree of T&D decreases the hidden information that offers private benefits to shareholders, thereby weakening their power. Hence, higher transparency and disclosure enhance a firm's financial distress. Amba (2014) also argues that increased transparency and disclosure lowers a firm's financial capabilities. Peters and Bagshaw (2014) and Akhtaruddin (2005) have argued that transparency and disclosure are not responsible for a firm's financial distress. The researchers have contrasting views on the connection between T&D and a firm's financial distress. Therefore, this study further augments the literature with new evidence on T&D and FD nexus in the Indian context. Hence, the following alternative hypothesis is framed.

**Hypothesis 1 (H1).** *Transparency and disclosure increase the firm's financial distress.*

Since the practice of financial liberalisation, competition has been raised for corporate sectors around the globe, especially in the third world. Investors seek the best opportunities that support their interests (Verrecchia 2001; Chau and Gray 2002). To win market share, it has become mandatory for domestic corporations to adopt a fully disclosed, transparent and uniform approach while reporting financial information. By this, the investor acts more sophisticatedly while deciding their business dealings. The reporting of financial information of corporations must meet internationally accepted practices. Modern technology and innovations in the corporate world have also raised the challenge of the tough competitive environment.

Verrecchia (2001) argues that a higher degree of transparency and disclosure is not advantageous for a firm's performance in this era of competition. More information available to competitors may decrease the market power of the firm. On a similar note, Farhi et al. (2013) have warned that higher revealing product information benefits the competitors, which may result in the low market power of the firm. The role of competitive advantage or disadvantage can not be avoided concerning the T&D of information. Hence, it would be interesting to find the moderating effect of competition on the association of T&D and FD. Thus, this study assumes the following alternative hypothesis (the conceptual model of the study is demonstrated in Figure 1):

**Hypothesis 2 (H2).** *Transparency and disclosure increase the firm's financial distress under higher competition.*

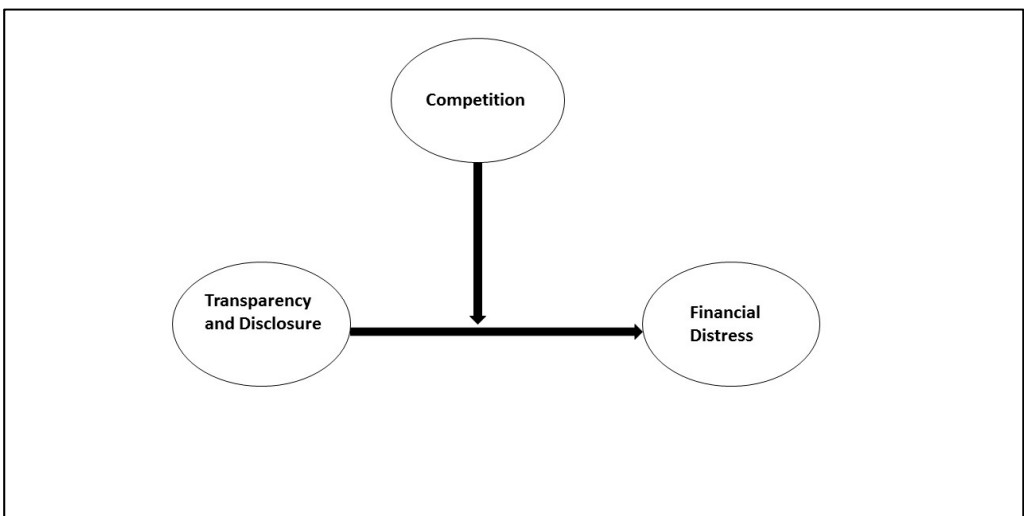

**Figure 1.** Conceptual Model.

### 3. Data and Methodology

*3.1. Data*

A dataset of 78 non-financial firms listed in BSE 100 indexed firms in India is taken for the study. We have excluded financial firms due to their different approach to information disclosures. Additionally, we have included the five financial years timeframe from 2015–2016 to 2019–2020. The rationale behind choosing this time frame is its importance due to recent regulatory reforms and, most importantly, the implementation of the Insolvency & Bankruptcy Code 2016 in India. In addition, the period after the financial year 2019–2020 is not considered because data were not adequately available due to the COVID-19 pandemic. The inclusion of COVID-19 and the later-on period might deliver inconsistent results due to missing data and the distraction from the normal course of action of firms' operations. The data is sourced from the CMIE prowess database and web portal of BSE (Bombay Stock Exchange). To have fine data, we have only considered 78 firms out of 100.

*3.2. Methodology*

The present study employs panel data econometrics for analysis. This approach gives more information, including the features of both time series and cross-sections (Hsiao 2007; Wooldridge 2015). Hence, panel data models are less prone to endogeneity issues and deliver unbiased results with more information. Three versions of the dependent variable (i.e., financial distress) are taken for model specification. Further, we have tested the relationship of T&D and FD under three conditions: (1) linear connectivity between T&D and FD, (2) nonlinear (quadratic) relationship, and (3) impact of T&D on FD under

competition. Moreover, both static and dynamic models are run to ensure the robustness of the results. Therefore, we have established a total of 18 ($3 \times 3 \times 2 = 18$) models. All the fundamental models are specified below:

Static Models:

$$DV_{it} = \beta_1 T\&D_{it} + \beta_2 LI_{it} + \beta_3 Opmargin_{it} + \beta_4 l\_sales_{it} + \beta_5 l\_mcap_{it} + u_{it} \tag{1}$$

$$DV_{it} = \beta_1 T\&D2_{it} + \beta_2 LI_{it} + \beta_3 Opmargin_{it} + \beta_4 l\_sales_{it} + \beta_5 l\_mcap_{it} + u_{it} \tag{2}$$

$$DV_{it} = \beta_1 T\&D_{it} + \beta_2 T\&D\_LI_{it} + \beta_3 Opmargin_{it} + \beta_4 l\_sales_{it} + \beta_5 l\_mcap_{it} + u_{it} \tag{3}$$

Dynamic Models:

$$DV_{it} = \beta_0 DV_{it\,(-1)} + \beta_1 T\&D_{it} + \beta_2 LI_{it} + \beta_3 Opmargin_{it} + \beta_4 l\_sales_{it} + \beta_5 l\_mcap_{it} + u_{it} \tag{4}$$

$$DV_{it} = \beta_0 DV_{it\,(-1)} + \beta_1 T\&D2_{it} + \beta_2 LI_{it} + \beta_3 Opmargin_{it} + \beta_4 l\_sales_{it} + \beta_5 l\_mcap_{it} + u_{it} \tag{5}$$

$$DV_{it} = \beta_1 DV_{it\,(-1)} + \beta_1 T\&D_{it} + \beta_2 T\&D\_LI_{it} + \beta_3 Opmargin_{it} + \beta_4 l\_sales_{it} + \beta_5 l\_mcap_{it} + u_{it} \tag{6}$$

where, $u_{it} = \mu_{it} + \nu_{it}$

The coefficients are indicated by $\beta_j$. DV shows the dependent variable, financial distress (FD). Z1, Z2, and Z3 are the three variants used for FD. 'T&D' and 'LI' are explanatory variables showing the T&D and competition levels, respectively. Opmargin, l_sales, and l_mcap are controlled variables. 'T&D2' is the squared term of 'T&D' for the nonlinear association. T&D_LI (T&D*LI) is an interaction term to explore the impact of T&D under competition. To include $\mu_{it}$ (individual-effect) and $\nu_{it}$ (regular error-term), $u_{it}$ is used. 'it' shows firm '$i$' at time $t$. '$-1$' is for lagged value. The variables are discussed in Table 1.

**Table 1.** List of Variables.

| Variable | Type | Symbol | Particular | Reference |
|---|---|---|---|---|
| Financial Distress (Altman Zscore) | DV | Z1 | It is the measure of firm's FD based on Altman FD model (as discussed in Appendix A.1) | Altman (1968), Shahwan (2015), and Pradhan (2014) |
| Financial Distress (BOS Score) | DV | Z2 | It is another measure of FD having Investors' Perspective. (Please see Appendix A.1 for detail) | Berger et al. (1996), Al-Hadi et al. (2017) |
| Financial Distress (AC Score) | DV | Z3 | It is an updated version of BOS Score model. (Please see Appendix A.1 for detail) | Almeida and Campello (2007), Al-Hadi et al. (2017) |
| Transparency and disclosure (T&D) | EV | T&D | It is the index generated for quantifying level of Transparency and disclosure. (Please see the detail in Appendix A.2) | Aksu and Kosedag (2006), Kumar and Kidwai (2018) |
| Competition (Lerner Index) | EV | LI | It is the computation of market competitiveness. (Please see the detail in Appendix A.3) | Lerner (1934), Sharma (2011) |

**Table 1.** *Cont.*

| Variable | Type | Symbol | Particular | Reference |
|---|---|---|---|---|
| Operational margin | CV | Opmargin | It signifies the profitability of the firm. It is calculated as dividing operational profit by net income. | Barth et al. (1998) |
| Market Capitalization (mcap) | CV | l_mcap | It represents the firm's value. The log value of mcap is taken. | Barth et al. (1998) |
| Sales | CV | l_sales | It also shows the firm's value. The log value of sales is taken. | Barth et al. (1998) |

*Note:* DV shows dependent variable. EV is for explanatory variable. CV is control variable.

### 3.3. Variables

The study mainly used six variables, as described in Table 1. Financial distress is the dependent variable. Three proxies of financial distress (Altman Zscore (as Z1), BOS Score (as Z2), and AC Score (as Z3)) have been used (see Appendix A.1 for detail). Transparency and disclosure (T&D) is the primary explanatory variable (see Appendix A.2 for more information). Competition is used as moderating variable to look for the interaction effect of transparency and disclosure and financial distress under competition. Competition is approximated by LI (see Appendix A.3 for details). Opmargin is taken as the control variable (see Table 1). l_mcap and l_sales (indicating valuation) are other control variables used in the study (see Table 1).

## 4. Results

### 4.1. Descriptive Statistics and Multicollinearity

Table 2 presents the descriptive statistics of variables included in the current study. The average value of Z1 (Altman Zscore) is 14.09 showing that the NFFs listed in India are in the safe zone from financial distress (because this mean value comes under the safe zone as discussed by Altman (1968)). On considering the investors' perspective, Z2 (BOS Score) and Z3 (AC Score) have mean values of 0.550 and 0.558, respectively, indicating that NFFs in India may be in financial distress (as these values are close to MIN values of 0.003 and 0.025). T&D has a mean value of 0.595 and is slightly close to the MAX value of 0.779. This mean value of T&D shows a moderate level of information disclosure by firms. The average of LI is 0.206 showing high proximity to MIN. Hence, it shows a higher degree of competition among sample NFFs. The averages of l_mcap and l_sales are 10.77 mean value of transparency and disclosure shows a moderate level of information disclosure, and 9.59, respectively, which are slightly close to MAX values of 13.60 and 13.33, showing on average a good value of NFFs in India. The average value of opmargin is 0.133 (closer to MIN value −0.418), indicating a lower Opmargin in NFFs listed in India.

**Table 2.** Descriptive Statistics.

| Variable | Mean | Max | Min | SD |
|---|---|---|---|---|
| Z1 (Altman Zscore) | 14.098 | 44.941 | −38.898 | 41.302 |
| Z2 (BOS Score) | 0.550 | 24.537 | 0.003 | 2.196 |
| Z3 (AC Score) | 0.588 | 24.545 | 0.025 | 2.193 |
| T&D | 0.595 | 0.779 | 0.310 | 0.095 |
| LI | 0.206 | 0.973 | 0 | 0.134 |
| Opmargin | 0.133 | 0.987 | −0.418 | 0.116 |
| l_mcap | 10.770 | 13.609 | 7.717 | 0.982 |
| l_sales | 9.598 | 13.330 | 5.303 | 1.33 |

*Note:* Z1, Z2, and Z3 are Altman Z-score, BOS Score, and AC Score, respectively. l_sales and l_mcap are natural log of sales and market capitalisation. Max is maximum value. Min is minimum value. SD is standard deviation.

They are, furthermore, looking at the correlation matrix in Table 3. The maximum significant correlation is found between l_mcap and l_sales, having a coefficient of 0.549. The maximum correlation coefficient is quite lower than the threshold of 0.80. Therefore, there are no worries about multicollinearity between the variables (Wooldridge 2015).

**Table 3.** Correlation Matrix.

|  | Z1 | Z2 | Z3 | T&D | LI | Opmargin | l_mcap | l_sales | T&D2 | T&D*LI |
|---|---|---|---|---|---|---|---|---|---|---|
| Z1 | 1 |  |  |  |  |  |  |  |  |  |
| Z2 | 0.272 * | 1 |  |  |  |  |  |  |  |  |
| Z3 | 0.274 * | 0.999 * | 1 |  |  |  |  |  |  |  |
| T&D | 0.007 | −0.006 | −0.003 | 1 |  |  |  |  |  |  |
| LI | 0.021 | 0.097 | 0.098 | 0.042 | 1 |  |  |  |  |  |
| Opmargin | 0.196 | −0.053 | −0.052 | −0.035 | 0.381 * | 1 |  |  |  |  |
| l_mcap | 0.160 * | 0.070 | 0.072 | 0.088 | 0.086 | 0.199 * | 1 |  |  |  |
| l_sales | 0.021 | 0.131 * | 0.129 * | 0.167 * | −0.254 * | −0.235 * | 0.549 * | 1 |  |  |
| T&D2 | 0.002 | −0.014 | −0.011 | 0.994 * | 0.042 | −0.041 | 0.096 | 0.179 * | 1 |  |
| T&D*LI | 0.023 | −0.090 | −0.090 | 0.260 | 0.967 | 0.354 | 0.098 | −0.218 * | 0.260 * | 1 |

*Note:* * shows a significant correlation coefficient at 0.05.

### *4.2. Regression Analysis*

A total of 18 models are developed to examine the transparency and disclosure and financial distress relationship, including base (linear) models, nonlinear models, and interaction models, each for three proxies of financial distress (Z1, Z2, and Z3). Each model includes static and dynamic versions of the panel data model.

#### 4.2.1. Base Models

Tables 4 and 5 demonstrate the base model results. The motive of base models is to analyse the linear association of T&D and FD (including Z-score, BOS Score, and AC Score). Further, both static and dynamic panel data models are employed. Hence, a total of $3 \times 2$ base models are developed. Models 1, 2, and 3 are specified considering the static model (see Table 4). Applying the F-test for fixed effect in these static models exhibits significant values at 1% significance. However, Bruesh–Pagan (BP) test for random effect also shows significant values at 1% significance for all three models. Choosing the fixed or random effect for the models creates confusion. Therefore, the Hausman test is applied, resulting in an insignificant value of 10%. This result confirms that these three models are well-suitable for random effect (Baltagi et al. 2003; Wooldridge 2015). Furthermore, the Wald and the Wooldridge test show significant values at 5% significance. Therefore, it confirms the existence of heteroscedasticity and autocorrelation. Thus, the study considers robust estimates to discuss results (Wooldridge 2015). Models 4, 5, and 6 are associated with the dynamic model. The Sargan and Arellano–Bond tests ensure the non-availability of over-identification and autocorrelation in these dynamic models (Judson and Owen 1999).

**Table 4.** Base Models Result for Linear Relation (Static Panel Data Analysis).

|  | Model 1 DV: Z1 (RE) | | Model 2 DV: Z2 (RE) | | Model 3 DV: Z3 (RE) | |
|---|---|---|---|---|---|---|
|  | Normal | Robust | Normal | Robust | Normal | Robust |
| T&D (exp_var) | −7.705 | −7.705 | 0.702 | 0.702 | 0.807 | 0.807 |
| LI | −6.298 | −6.298 | −0.516 | −0.516 | −0.539 | −0.539 |
| Opmargin | 0.181 | 0.181 | −0.002 | −0.002 | −0.002 | −0.002 |
| l_sales | −2.521 | −2.521 | 0.114 | 0.114 | 0.101 | 0.101 |
| l_mcap | 5.307 * | 5.307 * | 0.130 | 0.130 | 0.139 | 0.139 |

**Table 4.** *Cont.*

| | Model 1 DV: Z1 (RE) | | Model 2 DV: Z2 (RE) | | Model 3 DV: Z3 (RE) | |
|---|---|---|---|---|---|---|
| | **Normal** | **Robust** | **Normal** | **Robust** | **Normal** | **Robust** |
| *Cons.* | −16.416 | −16.416 | −2.207 | −2.207 | −2.197 | −2.197 |
| F-test (Model) | 11.78 ** | | 3.20 | | 2.48 | |
| F-test (Fixed effect) | 43.44 * | | 16.39 * | | 16.34 * | |
| BP-test (Random effect) | 610.82 * | | 448.73 * | | 447.92 * | |
| Hausman Test | 3.60 | | 1.02 | | 1.06 | |
| Wald test for Heteroscedasticity [1] | $1.6 \times 10^8$ * | | $6.4 \times 10^8$ * | | $2.0 \times 10^8$ * | |
| Wooldridge Autocorrelation Test [2] AR (1) | 91.068 * | | $5.57 \times 10^6$ * | | $7.39 \times 10^6$ * | |
| Sigma_$u_i$ | 39.071 | | 1.966 | | 2.1470 | |
| Sigma_$v_i$ | 13.049 | | 1.083 | | 0.7997 | |
| rho | 0.899 | | 0.767 | | 0.8781 | |
| R-Square | 0.047 | | 0.020 | | 0.4221 | |

*Note:* [1] Wald test examines heteroscedasticity having "null of no heteroscedasticity". [2] Wooldridge test examines autocorrelation in panel having "null of no autocorrelation" (with 1 lag). BP (Bruesch–Pagan) test looks for random effect. Sigma_$u_i$ and Sigma_$v_i$, respectively, are variance of individual effect (firms in this case) and error-term. The rho is the fraction of variance due to ui. Robust estimates are projected as of significant heteroscedasticity and/or autocorrelation. Parenthesis has *p*-value. *, **, *** are significance at 0.01, 0.05, and 0.10, respectively. exp_var is main explanatory variable and LI is proxy for competition measured using the Lerner index. 'T&D' is the T&D index.

**Table 5.** Base Models for Linear Relationship (Dynamic Panel Data Analysis).

| | Model 4 DV: Z1 | Model 5 DV: Z2 | Model 6 DV: Z3 |
|---|---|---|---|
| | **Coeff.** | **Coeff.** | **Coeff.** |
| Lag (1) | 0.549 * | 0.004 | 0.026 |
| T&D (exp_var) | −16.365 *** | −1.170 * | −1.057 * |
| LI | −0.481 | −0.046 | −0.031 |
| Opmargin | 0.043 | 0.000 | 0.000. |
| l_sales | −2.569 | −0.046 | −0.049 |
| l_mcap | 2.345 *** | 0.011 | 0.017 |
| *Cons.* | 11.209 | −2.306 | 1.401 * |
| Sargan-Test | 5.477 | 42.90 * | 38.08 * |
| Arellano-Bond Test | 0.317 | 1.01 | 1.00 * |

*Note:* Sargan test examines over identification problem under GMM framework having "null hypothesis of no over-identification problem" in dynamic panel data model. Arnello–Bond test used in the analysis is for serial autocorrelation in the first differenced error terms of the order 1. The null hypothesis of the test is "there is no autocorrelation". *, **, *** are significance at 0.01, 0.05, and 0.10, respectively. exp_var is main explanatory variable. LI is the proxy of competition measured using the Lerner index. 'T&D' is the T&D index. Lag(1) is lag of dependent variable at order 1. Coeff. is regression equation's coefficient value.

In all the static base models (Table 4), T&D and LI do not show any significant coefficient. This result indicates that T&D and competition do not impact a firm's FD. Among the control variables, only l_mcap (market capitalisation) is found significant and positive (with a coefficient of 5.307 and *p*-value of 0.005 < 0.05 in Model 1).

In dynamic models (Model 4, 5, and 6 in Table 5), the lag distress scores Z1(−1) and Z3(−1) in Model 4 and 6 have significant and positive coefficients (value 0.549 in Model 4 and 0.026 in Model 6) indicating the previous status of firm's financial stability positively impacts the present condition of financial stability. However, Z2(−1) in Model 5 does not exhibit a significant coefficient. T&D shows negative and significant coefficients for all distress

scores. 'T&D' is negative (−16.365) and significant for Zscore (Z1). The T&D is also negative and significant for BOS (Z2) and AC (Z3) scores with coefficients of −1.170 and −1.057, respectively. This situation implies that a higher degree of T&D increases a firm's financial distress (or reduces financial stability). Like static base models, LI does not show significant coefficients for FD, and l_macp is found to be a positive and significant control variable.

### 4.2.2. Nonlinear Models

For the investigation of the nonlinear association between T&D and the firm's FD, Models 7, 8, and 9 (static Models) and Models 10, 11, and 12 (dynamic Models) are developed. The results of nonlinear models are shown in Tables 6 and 7 for the static and dynamic models. Similar to the previous static models, F-test for fixed effect in the static models (Models 7, 8, and 9) exhibits a significant value. However, the Breush–Pagan test for random effect also shows significant values for all three models. Therefore, the Hausman test is applied, resulting in an insignificant value. It confirms the consistency of Models 7, 8, and 9 with random effect (Baltagi et al. 2003; Wooldridge 2015). Furthermore, the Wald test (testing heteroscedasticity) and the Wooldridge test (testing autocorrelation) show significant values. Therefore, it confirms the existence of heteroscedasticity and autocorrelation. Thus, the study considers robust estimates to discuss results (Wooldridge 2015). Sargan and Arellano–Bond tests report no overidentification and autocorrelation in dynamic models (Judson and Owen 1999).

**Table 6.** Static Models for Nonlinear Relationship.

| | Model 7 DV: Z1 (RE) | | Model 8 DV: Z2 (RE) | | Model 9 DV: Z3 (RE) | |
|---|---|---|---|---|---|---|
| | Normal | Robust | Normal | Robust | Normal | Robust |
| T&D2 (exp_var) | −6.392 (0.659) | −6.392 (0.513) | 0.459 (0.678) | 0.459 (0.267) | 0.543 (0.623) | 0.543 (0.189) |
| LI | −6.296 (0.545) | −6.296 (0.233) | −0.507 (0.543) | −0.507 (0.327) | −0.530 (0.525) | −0.530 (0.305) |
| Opmargin | 0.182 (0.143) | 0.182 (0.257) | −0.002 (0.794) | −0.002 (0.367) | −0.002 (0.770) | −0.002 (0.310) |
| l_sales | −2.497 (0.249) | −2.497 (0.282) | 0.118 (0.476) | 0.118 (0.294) | 0.105 (0.526) | 0.105 (0.351) |
| l_mcap | 5.300 * (0.003) | 5.300 * (0.005) | 0.129 (0.351) | 0.129 (0.352) | 0.138 (0.316) | 0.138 (0.317) |
| *Cons.* | −18.852 (0.482) | −18.852 (0.370) | −1.986 (0.255) | −1.986 (0.389) | −1.943 (0.265) | −1.943 (0.399) |
| F-test (Model) F-test (Fixed effect) | 11.80 ** (0.037) 43.44 * (0.000) | | 3.11 (0.683) 16.37 * (0.000) | | 3.23 (0.664) 16.32 * (0.000) | |
| BP-test (Random effect) | 610.83 * (0.000) | | 448.20 * (0.000) | | 447.40 * (0.000) | |
| Hausman Test | 3.57 (0.613) | | 1.01 (0.961) | | 1.05 (0.958) | |
| Wald test for Heteroscedasticity [1] | $1.3 \times 10^8$ * (0.000) | | $1.9 \times 10^8$ * (0.000) | | $2.0 \times 10^9$ * (0.000) | |
| Wooldridge Autocorrelation Test [2] AR (1) | 91.169 * (0.000) | | $5.89 \times 10^6$ * (0.000) | | $7.76 \times 10^6$ * (0.000) | |
| Sigma_$u_i$ | 39.072 | | 1.965 | | 1.962 | |
| Sigma_$v_i$ | 13.049 | | 1.084 | | 1.084 | |
| rho | 0.899 | | 0.766 | | 0.766 | |
| R-Square | 0.047 | | 0.020 | | 0.020 | |

*Note:* As mentioned in Table 4. Additionally, T&D2 is square of 'T&D' for nonlinear association.

**Table 7.** Nonlinear Models (Dynamic Panel Data Analysis).

| | Model 10 DV: Z1 | Model 11 DV: Z2 | Model 12 DV: Z3 |
|---|---|---|---|
| | **Coeff.** | **Coeff.** | **Coeff.** |
| Lag (1) | 0.540 * | 0.001 | 0.021 ** |
| T&D2 (exp_var) | −11.060 | −0.829 * | −0.746 * |
| LI | −0.356 | −0.039 | −0.030 |
| Opmargin | 0.048 | 0.000 | 0.000 |
| l_sales | −2.960 | −0.041 | −0.041 |
| l_mcap | 2.404 *** | 0.013 | 0.020 |
| *Cons.* | 9.080 | 0.995 * | 0.914 ** |
| Sargan Test | 5.412 | 37.36 * | 33.31 * |
| Arellano–Bond Test | 0.338 | 1.01 | 1.00 |

*Note:* As mentioned in Table 5. Additionally, T&D2 is square of 'T&D' for nonlinear association.

In static models (Models 7, 8, and 9 in Table 6), T&D2 (square of 'T&D') does not show significant coefficients. LI also has insignificant coefficients. However, only l_mcap is found significant for Z1 with a coefficient of 5.300 in Model 7.

For dynamic models (see Table 7), lag distress scores (Z1(−1) and Z3(−1)) are found significant and positive with coefficients 0.540 and 0.021, respectively, in Models 10 and 12. This result implies that the previous financial stability level increases the firms' current level of financial stability. Model 11 does not exhibit a significant coefficient for Z2(−1). In Models 11 and 12, 'T&D2' shows negative and significant coefficients (−0.829 and −0.746) for Z2 and Z3, respectively. This result indicates an inverted U shape relation between T&D and the firm's financial stability (the higher the distress score, the lower the FD). This result implies that a higher level of T&D initially increases a firm's financial stability (or decreases FD). However, it reduces the firm's financial stability (or increases FD) after a maximum threshold. In Model 10, 'T&D2' does not have a significant coefficient for Z1. LI does not show significant coefficients for distress scores in Models 10, 11, and 12. Again, only l_mcap as the control variable is found significant and positive with a coefficient of 2.404 in Model 10.

### 4.2.3. Interaction Models

There are six models for investigating the association of T&D and FD (Z1, Z2, and Z3) under the interaction of competition. Models 13, 14, and 15 align with the static model setup (see Table 8). Models 16, 17, and 18 are associated with a dynamic model framework (Table 9). As per previous static models (Models 1, 2, 3,7, 8, and 9), similar diagnostics are found for Models 13, 14, and 15 (static models). The random effect is followed in all models (Baltagi et al. 2003; Wooldridge 2015). Robust estimates are reported (Wooldridge 2015) due to autocorrelation and heteroscedasticity in static models. Sargan and Arellano-Bond tests confirm the non-availability of over-identification and autocorrelation, respectively (Judson and Owen 1999) in the dynamic models (Models 16, 17, and 18).

In static Models (Table 8), the explanatory variable 'T&D' (transparency and disclosure) has no significant coefficients. Additionally, the coefficients of T&D_LI (Interaction term = T&D*LI) are insignificant. The coefficients of LI in each model (Models 13, 14, and 15) are not significant. Only l_mcap is positive (5.330) and a significant control variable for Z1 in Model 13.

In Table 9, Models 16, 17, and 18 are associated with the dynamic model. Here also, the lag distress scores Z1(−1) and Z3(−1) in Models 16 and 18 have significant and positive coefficients (0.549 and 0.026), indicating the previous status of financial stability positively impacts the present condition of financial stability. However, Z2(−1) in Model 17 does not exhibit a significant coefficient. 'T&D' shows similar outcomes discussed in the earlier models (negative and significant coefficients). The coefficients of T&D_LI (Interaction term = T&D*LI) are also insignificant. It indicates that competition does not significantly affect

the association of transparency and disclosure (T&D) and financial distress. LI does not have significant coefficients. Again, only l_mcap as a control variable is significant and positive in Model 16.

**Table 8.** Interaction Models (Static Panel Data Analysis).

| | Model 13 DV: Z1 (RE) | | Model 14 DV: Z2 (RE) | | Model 15 DV: Z3 (RE) | |
|---|---|---|---|---|---|---|
| | Normal | Robust | Normal | Robust | Normal | Robust |
| T&D (exp_var) | −5.414 | −5.414 | 0.871 | 0.871 | 0.983 | 0.983 |
| T&D_LI | −10.772 | −10.772 | −0.828 | −0.828 | −0.861 | −0.861 |
| Opmargin | 0.184 | 0.184 | −0.002 | −0.002 | −0.002 | −0.002 |
| l_sales | −2.553 | −2.553 | 0.113 | 0.113 | 0.100 | 0.100 |
| l_mcap | 5.330 * | 5.330 * | 0.130 | 0.130 | 0.140 | 0.140 |
| *Cons.* | −17.745 | −17.745 | −2.306 | −2.306 | −2.298 | −2.298 |
| F-test (Model) F-test (FE) | 11.84 ** 43.47 * | | 3.21 16.41 * | | 3.34 16.36 * | |
| BP-test (RE) | 611.54 * | | 449.30 * | | 448.52 * | |
| Hausman Test | 3.49 | | 0.95 | | 0.99 | |
| Wald test for Heteroscedasticity [1] | $2.5 \times 10^8$ * | | $3.7 \times 10^9$ * | | $3.0 \times 10^8$ * | |
| Wooldridge Autocorrelation Test [2] AR (1) | 90.993 * | | $5.4 \times 10^6$ * | | $7.4 \times 10^6$ * | |
| Sigma_$u_i$ | 39.094 | | 1.968 | | 1.965 | |
| Sigma_$v_i$ | 13.048 | | 1.083 | | 1.084 | |
| rho | 0.899 | | 0.767 | | 0.766 | |
| R-Square | 0.047 | | 0.019 | | 0.019 | |

*Note:* As mentioned in Table 4. Additionally, T&D_LI (=T&D*LI) represents interaction variable having LI as moderator.

**Table 9.** Interaction Models (Dynamics Panel Data Analysis).

| | Model 16 DV: Z1 | Model 17 DV: Z2 | Model 18 DV: Z3 |
|---|---|---|---|
| | Coeff. | Coeff. | Coeff. |
| Lag (1) | 0.549 * | 0.004 | 0.026 * |
| T&D (exp_var) | −16.125 *** | −1.165 * | −1.066 * |
| T&D_LI | −1.010 | 0.005 | 0.045 |
| Opmargin | 0.043 | 0.000 | 0.000 |
| l_sales | −2.569 | −0.048 | −0.050 |
| l_mcap | 2.399 *** | 0.011 | 0.016 |
| *Cons.* | 11.023 | 1.511 * | 1.416 * |
| Sargan Test | 5.439 | 43.36 * | 38.60 * |
| Arellano–Bond Test | 0.317 | 1.01 | 1.00 |

*Note:* As mentioned in Table 5. Additionally, T&D_LI (=T&D*LI) represents interaction variable having LI as moderator.

### 4.3. Endogeneity and Robustness Check

This study also tests the endogeneity issues arising from the main explanatory variables (Wooldridge 2015; Kanoujiya et al. 2022). Applying the Wu–Hausman test and the Durbin–Watson Chi2 test, the exhibited p-values by these tests are not significant (higher than 0.50) except for the control variable l_sales (Table 10). Therefore, the issue of endogeneity due to

explanatory variables for dependent variables is discarded (Wooldridge 2015; Kanoujiya et al. 2022). The third lag of the independent variables is used as an instrument variable (IV) in the regression for testing endogeneity (Wooldridge 2015; Kanoujiya et al. 2022).

**Table 10.** Endogeneity Test.

|  | **T&D** | **LI** | **Opmargin** | **l_mcap** | **l_sales** | **T&D2** | **T&D_LI** |
|---|---|---|---|---|---|---|---|
| sDurbin Chi-2 | 1.8063 (0.1789) | 0.0222 (0.8688) | 2.5703 (0.1089) | 0.3633 (0.5462) | 5.7976 * (0.0160) | 2.2285 (0.1355) | 0.0041 (0.9488) |
| Wu–Hausman Test | 1.7426 (0.1889) | 0.02601 (0.8722) | 2.4926 (0.1166) | 0.3467 (0.5564) | 5.7489 * (0.0178) | 2.1561 (0.1442) | 0.0039 (0.9501) |

*Note: p*-value is shown in (). * shows a significant value at 5% significance level.

The present study employs a multi-method approach, including several variants of the dependent variable (Z1, Z2, and Z3) to ensure the robustness of the results (Kanoujiya et al. 2022). In addition, both static and dynamic models are employed. In most cases, similar outcomes are found, indicating that T&D increases the firm's FD. Moreover, for the interaction effect, no model reveals any significant link between T&D and FD under competition. The similarities in the results ensure the results' robustness.

## 5. Discussion

### 5.1. Hypothesis Discussion

In all dynamic models, enough evidence is shown for the association of T&D with the firm's FD (including both linear and nonlinear association). Therefore, hypothesis H1 cannot be denied. This situation implies that a higher degree of T&D amplifies the firm's FD (reduces financial stability). Moreover, it is also evident from the quadratic (nonlinear) relationship that the T&D level increases the financial stability to a certain upper limit. Then it adversely impacts the firm's financial stability beyond that limit (inverted U-shaped relation). In other words, initially, T&D decreases the firm's FD to a minimum point, and after that, it increases FD. However, Static models do not exhibit evidence for the support of H1. No significant evidence is found for the interaction effect from any model. This outcome supports hypothesis H2. Therefore, this implies that T&D does not matter for a firm's FD under the pressure of competition.

### 5.2. Results' Comparison with Previous Findings in the Literature

The studies on the transparency and disclosure and firm financial distress relationship are very limited in number. However, the current findings are supportive of the conclusions from Berglöf and Pajuste (2005) and Verrecchia (2001), James-Overheu and Cotter (2009) and Hermalin and Weisbach (2012). They argue that higher T&D is detrimental to a firm's financial stability and increases FD. However, the current findings are not in support of Lang et al. (2012), Östberg (2006), Bebchuk et al. (2009), Chau and Gray (2002), Amba (2014) and Lai et al. (2014). They advocate that increasing T&D improves a firm's financial health and lowers financial distress. The current findings are also not in line with Hussin and Othman (2012), Abdul-Qadir and Kwanbo (2012), and Cheffins (2013). They, too, indicate a negative connection of T&D to the firm's FD. Furthermore, the current findings are contradictory to the findings of Ahmed and Courtis (1999) and Akhtaruddin (2005) and Peters and Bagshaw (2014), which find no significant association of T&D with FD.

### 5.3. Contribution

The present era of India's corporate sector is quite different from the pre-reform era. T&D is being upheld at present, while it was elusive before. (Thapar and Sharma 2017). India's corporate world has witnessed many structural changes as it is embroiled in local and global competition (Balasubramanian 2013; Thapar and Sharma 2017). With recent amendments in the Companies Acts and the introduction of the Insolvency and Bankruptcy

Code (IBC) 2016, it has become a critical issue to provide new evidence on T&D and its connection to a firm's financial stability. The study's findings demonstrate that T&D influences a firm's FD. Therefore, in this vein, the current study contributes to the available literature on T&D and FD by providing fresh evidence on the association between T&D and FD of non-financial financial firms in India. Moreover, the study has investigated the nonlinear relationship and linear relation of T&D to a firm's FD under the interaction effect of competition.

Furthermore, this study has been done considering several dimensions of a firm's health, which includes Altman Z-score in general and two other measures for financial stability. Thus, no such study is available in the literature to the authors' knowledge. Hence, the current paper is unique research of its kind, particularly on India's listed NFFs.

*5.4. Implications*

T&D plays a vital role in a firm's CG practices. FD is a critical issue that must be handled promptly (Fama and Jensen 1983; Mallin 2016). In this vein, T&D helps to have informed decisions by stakeholders, shareholders, investors, and policymakers regarding the firm's financial health (Fung 2014; Chi et al. 2009).

The current study's findings have many noteworthy and insightful implications for managers, policymakers, and investors. First, T&D adversely impacts financial stability. Hence managers should give proper attention while disclosing information. Second, the level of T&D should not go beyond a certain upper limit, as T&D first decreases FD and then increases FD. In other way, initially, T&D helps in improving financial stability. Then it lowers financial stability after a maximum limit (U shape for financial distress and inverted U shape for financial stability). Under competition, T&D has nothing to do with the firm's FD. This result means T&D can be compromised under competitive market conditions. For the policy makers' managers, it is essential to consider all dimensions of T&D in framing rules and regulations. They should keep a suitable T&D level to balance its benefits and costs. Most importantly, investors should give T&D due consideration as it is critical for a firm's financial stability.

## 6. Conclusions

T&D is an essential element for a strong CG, affecting the firms' financial stability (Fung 2014; Chi et al. 2009). The paper investigates the connection between the T&D and FD of NFFs listed in BSE India. The paper first assesses the FD and T&D of Indian NFFs. Secondly, it investigates the linear connection of T&D to the firm's FD. Thirdly, it also examines the nonlinear association of T&D with FD. Finally, the current study investigates the T&D's impact on a firm's FD under the interaction of competition. In most models, a significant connection between T&D and FD is found (including both linear and nonlinear establishment). These associations are negative, signalling that a higher level of T&D is not good for a firm's financial health.

Furthermore, the level of T&D should not go beyond a specific upper limit (U shape establishment for FD and inverted U shape for financial stability). However, it is also evident that T&D does not influence FD under the interaction of competition. Hence, the findings provide insights for all stakeholders, shareholders, and potential investors to decide on the firm's financial stability and capital allocation. This study is novel in its approach as it incorporates the firm's financial health in different dimensions, including investors' perspectives.

The current paper is not independent of its limitations. It should not be taken in general as its scope is limited to the non-financial firms listed in India. It does not look for financial firms due to their distinctive reporting strategy. The corporate world's political structure and regulatory frameworks vary from nation to nation. Hence, the current findings cannot be generalised for all nations. However, the authors believe it may provide insights into non-financial firms in similar economies. Moreover, the COVID-19 pandemic, wars and other events might affect business operations' normal course of action. These

events are not considered in the current investigation. Thus, owing to the limitations, this study recommends considering financial firms for the impact of T&D on FD in future studies. The studies on the current topic can be extended to examine the impact of T&D on FD in pre- and post-period of COVID-19, wars, or other such events. The current study should also be put forward to explore other economies incorporating more suitable features of corporate governance. A more accurate T&D index should be generated, including more contemporary constituents related to the concerned economy in the modern era. As a firm's financial distress is a critical issue, more factors responsible for financial distress need to be investigated.

**Author Contributions:** Conceptualization, S.R.; Methodology, R.A. and J.K.; Software, S.R.; Validation, V.M.B. and R.A.; Formal Analysis, S.R.; Investigation, J.K.; Resources, S.R.; Data Curation, V.M.B. and R.A.; Writing—Original Draft Preparation, J.K.; Writing, Review and Editing, S.R. and R.A.; Visualization, V.M.B.; Supervision, S.R., J.K. and R.A.; Project Administration, R.A.; Funding Acquisition, N/A. All authors have read and agreed to the published version of the manuscript.

**Funding:** This research received no external funding.

**Data Availability Statement:** Data is available on request to first author.

**Conflicts of Interest:** The authors declare no conflict of interest.

## Appendix A

### Appendix A.1. Financial Distress

Firm's FD is a situation when a firm's is incapable of meeting its financial obligations (Altman 1968; Kaur 2019). It may lead to unwanted consequences of bankruptcy and business failure. FD is used as dependent variable and is proxied by Altman Z-score, BOS distress score, and AC distress score.

*Altman Zscore:* Recognising the five most significant ratios (financial) out of 22 ratios, Altman (1968) has developed a model for predicting firm's FD. He has taken a sample of 66 firms (33 matching and 33 failed firms during 1946–1965) for his study. His model is the most acceptable model for measuring FD as of having 80–90% accuracy of the model (Altman 1968; Kaur 2019). The value obtained by the model is known as *Altman Z-score* (the higher the value the lower the FD). Based on multivariate discriminant analysis, the Altman Zscore model is given as:

$$Z\text{-}score\ (Z1) = 1.2xS1 + 1.4xS2 + 3.3xS3 + 0.6xS4 + 1.0xS5 \tag{A1}$$

where:

'$x$' is for multiplication.
$S1$ = working capital/total assets.
$S2$ = retained earnings/total assets.
$S3$ = EBIT/total assets.
$S4$ = market value of equity/book value of total liabilities.
$S5$ = sales/total assets.

Classification of companies are:

$Z1 > 2.67$—safer zone (No risk of FD)
$1.81 < Z1 < 2.67$—moderate zone (Prone to FD risk)
$Z1 < 1.81$—distressed zone (facing FD)

*BOS Score:* Berger et al. (1996) have developed another FD model with investors' opinion (Al-Hadi et al. 2017). The receivables (REC), inventory (INV), Net PPE (property plant and Equipment), and total assets (TA) are employed as model inputs. On their names, it is popularly known as BOS model to quantify firm's FD. The specified model is:

$$BOS\_Dis(Z2) = (0.715 * REC + 0.547 * INV + 0.535 * Net\ PPE)/TA) \tag{A2}$$

*AC Score:* For a more accurate model, Almeida and Campello (2007) amended the existing BOS model (Berger et al. 1996). They have included the Cash variable in model input to estimate more accurate results (Al-Hadi et al. 2017). The amended model is known as AC model of quantifying FD and is calculated as:

$$AC\_Dis(Z3) = ((Cash + 0.715 * REC + 0.547 * INV + 0.535 * Net\ PPE)/Total\ Assets) \quad (A3)$$

A higher level of model output represents a lower level of the firm's FD.

### *Appendix A.2. Transparency and Disclosure*

Following Aksu and Kosedag (2006), Kumar and Kidwai (2018) and Arsov and Bucevska (2017), this study has constructed a T&D index for the computation of T&D. S&P study had been taken into light and it is customised for constructing the T&D index. The study uses a total of 102 worthwhile T&D features that are extensively used in many countries. To make an effective T&D index model, the newer set of features (Strategic, Technology and Internet Disclosures) are also included.

The current T&D index model includes the following category of information:

1.  Financial Transparency and Information Disclosure (30 attributes),
2.  Board & Management Structures & Processes (29 attributes),
3.  Ownership Structure & Investor Relations (10 attributes) and
4.  Strategic, Technology, and Internet Disclosures (33 attributes).

The study follows the unweighted disclosure approach for building index as discussed by Arsov and Bucevska (2017) and Kumar and Kidwai (2018). Using binary form, '1' is valued for availability of information and '0' for non-availability.

### *Appendix A.3. Lerner Index*

For evaluation of competition, The Lerner Index (Lerner 1934) is utilised as discussed in Praveena and Samsai (2014). To evaluate the competition level, we have employed the following equation:

$$Liit = (Pit - MCit)/Pit$$

where, P indicates the net profit. MC is marginal cost taken as the firm's operating cost (Praveena and Samsai 2014). LI represents the Lerner Index. The higher value of LI signals stronger market power or a low level of competition. 'it' is for bank' i' and time 't'.

As the samples firms includes firm from several industries, hence industry specific factor should be adjusted (Sharma 2011). Therefore, an amended version of LI is applied in the current study (Sharma 2011):

$$LI_{IA} = LI_i - \sum_{i=1}^{N} \omega_i LI_i \quad (A4)$$

where $LI_{IA}$ is industry-adjusted LI, and $LI_i$ is Lerner Index of firm 'i'. $\omega_i$ indicates proportion of sales of firm *i* to total sales of the industry. A lower value shows a high competition.

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
