# Peer review of "Transparency and Disclosure and Financial Distress of Non-Financial Firms in India under Competition: Investors’ Perspective"

_jrfm, doi:10.3390/jrfm16040217_

Round 1

Reviewer 1 Report

The article was written without complying with the requirements for technical design and structure. A lot of outdated literature is also used. Accordingly, the substantiation of the hypotheses was carried out without sufficient consideration of modern realities and requirements for transparency of information and reporting, both financial and integrated. There is no analysis of the latest trends and reporting standards. The survey was conducted during 2016-2020, however, the conditions of NGO functioning, priorities, and tools of their activity have also changed significantly over the past couple of years. Significant changes are taking place in the world, including the covid pandemic and wars, which significantly change the operating conditions. The idea of the article is still promising, but its implementation needs significant improvement.

Author Response

See attached file for the changes made in the document.

Reviewer 2 Report

Research Article: “Transparency and Disclosure and Financial Distress of Non-Financial Firms in India under Competition: Investors’ Perspective”.

The issue addressed in the paper “Transparency and Disclosure and Financial Distress of Non-Financial Firms in India under Competition: Investors’ Perspective” The Authors proposed an important and interesting topic, compatible with the scope of the journal. Moreover, the author (s) briefly discusses the link among the all variables. This study itself is a new contribution to the current literature. However, there are some shortcomings throughout the text that need to be addressed. I wish you all the best in your work.

The following suggestions/corrections/changes should be incorporated to improve the quality of the above titled research article:

11)    The title of the paper seems good however; it is suggested to rephrase the existing title and write another title that can best explain your current study. Similarly, abstract has some shortcomings. I am suggesting the author to revise the abstract and correct the grammatical and other small mistakes. It is high recommended to carefully check the abstract before next submission.

22)    Is it possible to clearly write the gap and novelty of your study in one paragraphs?

33)    Please also revise structure of the paper according to the journal template.

44)    In the literature review please follow the standard structure. I am suggesting to replace the conceptual diagram at the end of literature review.

55)    Beside, If possible please start your Literature review from a standard paragraph that link your Introduction with Literature.

66)    Literature review is also not update. Please cite some recent papers.

https://doi.org/10.1016/j.bir.2023.01.001

https://doi.org/10.1108/CG-05-2021-0169

https://www.mdpi.com/2071-1050/15/4/2990

https://ejournal.unsri.ac.id/index.php/ja/article/view/19695

77)     Is it possible for you to check the dataset that why these value has too much difference. (Mean 14.098, Max 449.419, and -38.898). Is any data is written wrong?

88)    Please shorten the explanation in the result section. Only write your main findings in few lines.

99)    Again please follow the journal standard structure. Specially end of the paper is not so good.

110) Last but not the least, it is suggested to proofread this publishable article that will improve its language and rectify other typos mistakes.        

Best of Luck!

Author Response

See attached file for changes made in the document for Reviewer 2.

Reviewer 3 Report

I would like to congratulate authors for this study, which aimed to assess the relationship between transparency and disclosure of financial information with financial distress seen in listed companies in India. The authors concluded that transparency and disclosure positively affects the companies' bottom line but only to a certain extent, thereafter it starts to affect negatively. The article added more insight into a previously studied topic, as it addressed a country with no research on the topic, as explained.

The title could be more suggestive if related to the objective of the study. The study objective is very clear, and the abstract provides a detailed explanation of the methodological approach used to reach the conclusions.

To improve the readability and attractiveness of the article, I suggest reducing acronyms (many are similar) and make the text very confusing without any scientific gains.

The references are up to date and cover the main research areas of the article.

In the Introduction, the research topic is well presented, and despite being supported by literature review, it seems to result in a too long text for this chapter; a greater objectivity and synthesis capacity is suggested. In fact, the Introduction, although rich in information, is too long and with extensive theoretical grounding to support the choice of topic and research theme and may cause distraction to the less attentive reader. Perhaps some of the ideas presented in the introduction could be better framed in the literature review.

Concerning methodology, the methods applied are justified, valid and reliable and all variables were adequately defined and measured. The authors described the methodology correctly, providing sufficient detail to replicate the study.

Regarding the presentation and discussion of results, all data are presented appropriately, and tables are relevant and clearly presented. The authors are clearly able to distinguish what is a statistically significant or a practically significant result. Although the discussion was carried out within the scope of the hypotheses tested, we believe that the discussion between the results obtained and the results found in previous studies (literature review) must be substantially improved, particularly because they will reinforce the legitimacy of the contributions that the authors intend to argue.

The conclusions are adequate and focus on the objective of the study, being minimally supported by literature. However, the authors are scarce in indicating the study limitations, and the proposal for future lines of research is too subtle at the end of the conclusion, so these two topics could be more explicit and elaborated.

Author Response

See attached file for changes made in the document for Reviewer 3.

Round 2

Reviewer 2 Report

Congratulations! You have passed my review. Thank you for making the necessary amendments. Please wait for the editor's final decision based on your revisions.

Thank you very much.